# Lowering blood pressure after acute intracerebral haemorrhage: protocol for a systematic review and meta-analysis using individual patient data from randomised controlled trials participating in the Blood Pressure in Acute Stroke Collaboration (BASC)

Tom J Moullaali,[1,2] Xia Wang,[2] Lisa J Woodhouse,[3] Zhe Kang Law,[3,4] Candice Delcourt,[2,5] Nikola Sprigg,[3,6] Kailash Krishnan,[3,6] Thompson G Robinson,[7,8] Joanna M Wardlaw,[1] Rustam Al-Shahi Salman,[1] Eivind Berge,[9] Else C Sandset,[10,11] Craig S Anderson,[2,12] Philip M Bath,[3,6] The BASC Investigators

For numbered affiliations see end of article.

**Correspondence to**
Professor Craig S Anderson;
canderson@georgeinstitute.org.au

## ABSTRACT

**Introduction** Conflicting results from multiple randomised trials indicate that the methods and effects of blood pressure (BP) reduction after acute intracerebral haemorrhage (ICH) are complex. The Blood pressure in Acute Stroke Collaboration is an international collaboration, which aims to determine the optimal management of BP after acute stroke including ICH.

**Methods and analysis** A systematic review will be undertaken according to the Preferred Reporting Items for Systematic review and Meta-Analysis of Individual Participant Data (IPD) guideline. A search of Cochrane Central Register of Controlled Trials, EMBASE and MEDLINE from inception will be conducted to identify randomised controlled trials of BP management in adults with acute spontaneous (non-traumatic) ICH enrolled within the first 7 days of symptom onset. Authors of studies that meet the inclusion criteria will be invited to share their IPD. The primary outcome will be functional outcome according to the modified Rankin Scale. Safety outcomes will be early neurological deterioration, symptomatic hypotension and serious adverse events. Secondary outcomes will include death and neuroradiological and haemodynamic variables. Meta-analyses of pooled IPD using the intention-to-treat dataset of included trials, including subgroup analyses to assess modification of the effects of BP lowering by time to treatment, treatment strategy and patient's demographic, clinical and prestroke neuroradiological characteristics.

**Ethics and dissemination** No new patient data will be collected nor is there any deviation from the original purposes of each study where ethical approvals were granted; therefore, further ethical approval is not required. Results will be reported in international peer-reviewed journals.

## Strengths and limitations of this study

► The Blood pressure in Acute Stroke Collaboration is an international collaboration with the prospective aim of pooling individual participant data (IPD) from randomised controlled trials (RCTs) of blood pressure (BP) management in acute stroke, including intracerebral haemorrhage (ICH).
► Meta-analysis of IPD is regarded as the gold standard for synthesising evidence from multiple RCTs.
► This study aims to collect sufficient data to determine the optimal management of BP in acute ICH and will facilitate comparisons across subgroups according to patient characteristics, time to treatment and treatment strategy.
► This study will identify associations of treatment-related haemodynamic parameters, which will inform future research.

**PROSPERO registration number** CRD42019141136.

## INTRODUCTION
### Rationale

Acute treatments proven to alter the prognosis of stroke due to spontaneous (non-traumatic) intracerebral haemorrhage (ICH) are limited. A 2014 Cochrane review that appraised the totality of evidence for blood pressure (BP) lowering within 1 week of acute stroke concluded that the supporting evidence was insufficient to make clear recommendations about this intervention,

and the findings were similar for ischaemic stroke and ICH.[1] In regard to ICH, several randomised controlled trials (RCTs) have focused on early intensive BP reduction and included participants with mild-to-moderate acute ICH in the hospital setting.[2–4] Other trials commenced BP lowering in the prehospital period before diagnostic brain imaging, and by their nature included a mixture of ischaemic stroke and ICH patients, including more severe cases of ICH.[5–7] Data from ICH cases are also available from RCTs, which used longer inclusion windows (up to 48 hours).[8–12] Furthermore, various agents and BP targets have been tested, and few trials considered prognostic implications of prestroke neuroimaging characteristics. Therefore, a substantial and varied body of evidence about BP lowering and outcome after acute ICH is available, but much of it is conflicting.[13]

The Blood pressure in Acute Stroke Collaboration (BASC) is an international collaboration, which aims to prospectively pool individual participant data (IPD) from RCTs of BP control after acute stroke, including stroke due to ICH.[14 15] IPD meta-analysis is considered the gold-standard for synthesising evidence from RCTs[16] and, in this context, provides added value to meta-analyses of aggregate data[17] by facilitating multi-variable analyses of treatment effects and subgroup analyses according to time to treatment, treatment strategy used and baseline characteristics including neurological severity and haematoma volume, adjusted for confounding factors.

### Primary objective

To determine the effect of BP lowering on clinical outcomes in patients with acute ICH and elevated BP.

### Secondary objectives

1. To determine the effect of BP lowering according to baseline patient clinical and neuroradiological characteristics, BP lowering strategy and timing of the intervention.
2. To determine the effect of BP lowering on radiological outcomes.
3. Where sufficient data are available, report associations of on-treatment haemodynamic parameters, adjusted for known confounders.

### METHODS AND ANALYSIS

This systematic review and IPD meta-analysis will be performed in accordance with the recommendations made by the methods group of the Cochrane Collaboration (http://ipdmamg.cochrane.org/resources) and the UK Medical Research Council Network of Hubs for Trials Methodology Research.[18] The protocol has been developed in accordance with the Preferred Reporting Items for Systematic reviews and Meta-Analyses of IPD checklist[19] and has been registered with the international prospective register of systematic reviews, PROSPERO.[20]

### Patient and public involvement

This protocol was developed without the involvement of patients or members of the public.

### Eligibility criteria

#### Study designs

RCTs of BP reduction during the acute phase (<7 days) of stroke will be included.

#### Participants

Adults (aged >18 years) with spontaneous ICH will be included. For trials including both ICH and ischaemic stroke, we will include only patients with ICH. Participants with secondary ICH (eg, due to trauma, tumour or vascular malformation) will be excluded. Uncertainty about an individual trial's eligibility criteria will be clarified with the appropriate investigator.

#### Interventions

Trials that involve interventions that lower BP including oral, sublingual, transdermal and intravenous agents and single or combination therapy will be considered.

#### Comparators

Control groups will typically be managed with placebo or according to contemporaneous guideline recommendations.

#### Outcomes

Trials that report data on death and functional outcome using the seven-level modified Rankin Scale (mRS, where scores range from 0=no symptoms to 5=severe disability and 6=death) at 90 days will be included. Where outcome data are not reported this way, the nearest time point or suitable surrogate will be considered. Data regarding other outcomes (survival, quality of life, cognitive function), serious adverse events (SAEs), haematoma characteristics (growth, final volume, mass effect, oedema) and haemodynamic measures will be sought.

#### Setting

There will be no restrictions on the type of setting.

#### Language

There will be no language restriction.

### Information sources

Trials will be sought using electronic searches of Cochrane Central Register of Controlled Trials, EMBASE and MEDLINE and in the reference lists of published systematic reviews and *ad hoc* reviews. Ongoing or unpublished trials will be identified using ClinicalTrials.gov and the WHO's International Clinical Trials Registry Platform.

### Search strategy

Searches of the above bibliographic databases will be conducted using a combination of search terms relevant to the proposed study (online supplementary material 1). Databases will be searched from inception to the present. The search strategy will be enhanced through searches of

the existing BASC reviews, the authors' publication databases and reference lists in identified articles.

## Selection process

All titles and abstracts will be screened against the eligibility criteria by TJM and ZKL. Any disagreements will be settled by an adjudicator, PMB. The principal investigators of eligible studies will be invited to participate in the collaboration.

## Data management
### Data sharing

Investigators of eligible studies will receive a written invitation to share their IPD. To ensure transparency, collaborators sharing data with BASC will be asked to sign a data transfer agreement for the predefined and appropriate use of their data according to this protocol.

### Data checking

Initial internal analyses will compare data from each trial with their published results to ensure that data are complete and transferred without error: this will include checking the primary outcome and all baseline variables or secondary outcomes relevant to the proposed analyses. The integrity of all data will be checked and any queries resolved with individual trial investigators.

### Data merging

Datasets obtained from collaborating studies will be combined to form a new master dataset, which will include a variable to indicate the original study.

### Confidentiality, data storage and handling

Each collaborator will deidentify their dataset to ensure no patient identifiable information is transferred. Data will be shared electronically and stored on password-protected, encrypted hard disks in a locked room, with daily backup facilitating disaster recovery.[15] Data will not be shared with anyone outside the collaborating group.

## Data items

Variables that will be requested from participating investigators are listed in online supplementary material 2, and can be outlined as follows:
- Trial information
- Demographics
- Medical history
- Medications at the time of admission
- Baseline clinical variables
- Baseline neuroimaging characteristics of the acute ICH and prestroke features (leukoaraiosis, atrophy, prior stroke lesions)
- BP treatment and all trial BP data
- Clinical and radiological outcome data

## Study outcomes
### Primary outcome

The preferred primary effect variable will be functional outcome defined by the ordinal distribution of mRS scores at the end of trial follow-up (usually 3 months).

### Secondary outcomes

(1) Death and dependency (3–6 on the mRS); (2) death or severe disability (4–6 on the mRS); (3) all-cause death; and (4) health-related quality of life (mobility, self-care, usual activities, pain or discomfort and anxiety or depression), as assessed with the use of the European Quality of Life–5 Dimensions questionnaire.

### Safety outcomes

(1) Early neurological deterioration (as defined by each individual trial); (2) symptomatic hypotension (as defined by each individual trial); and (3) other SAEs, as defined by trial, to include fatal, non-fatal and treatment-related SAEs (including renal SAEs).

### Radiological outcomes

(1) Haematoma growth, both absolute and proportional, will be studied where these data were collected by individual trials and is based on central measurements from semiautomated planimetric[21] or ABC/2[22] methods; (2) other imaging outcomes (eg, perihaematomal oedema) will be included in supplemental analyses.

### Subgroup analyses

Where adequate data are available, heterogeneity in the effect of BP lowering on outcomes will be assessed in the following subgroups to determine whether any effects of BP lowering are moderated by patient's characteristics and BP lowering treatment (agent, target, timing or place of delivery):
- Baseline characteristics: demographic (age, sex, region), clinical (stroke severity, baseline systolic BP, prestroke antihypertensive drug use, prestroke antithrombotic use) and radiological parameters (ICH volume, pre-stroke characteristics).
- BP lowering strategy: intervention/class-based treatment, BP target-based treatment.
- Timing of intervention: <2 hours, 2–6 hours, 6–48 hours and >48 hours after onset of ICH.
- Type of trial: prehospital versus hospital and ICH only versus mixed.

### Associations of calculated BP parameters and primary outcome

Once all data have been pooled, an assessment of the new dataset will be made with a view to use trial BP and heart rate measures to calculate important BP lowering haemodynamic parameters, a number of which have been associated with outcome after ICH.[23–26] The aim is to determine the prognostic significance of these calculated variables, adjusted for all known confounders, and present these data in secondary analyses.

## Risk of bias in individual studies

Two investigators (TJM and ZKL) will assess each included study for bias using the Cochrane Collaboration tool (http://methods.cochrane.org/bias/assessing-risk-bias-included-studies). A judgement of bias will be made according to six domains: selection, performance, detection, attrition, reporting and 'other' biases. Disagreements

will be resolved with discussion or by involving PMB if necessary. Any uncertainties about trial design, conduct or analysis methods will be clarified with the individual trial's investigators.

### Statistical analyses

Primary analyses will be performed using the intention-to-treat dataset from each trial with a one-stage approach. Patients without available data or where the above procedure for missing data cannot be applied will be excluded from these analyses.

Online supplementary material 3 contains table and figure shells that will be used to present our findings. In summary, descriptive analyses will be undertaken in order to identify key similarities and differences between trials, therefore providing context for interpretation of between-trial differences in outcomes. Data will be described as mean (SD) or median (IQR) for continuous data or frequency (percentage) for categorical data, and chi-squared or Kruskal-Wallis tests will be used to make comparisons.

Generalised linear mixed models will be used with covariates (age, sex, ICH severity, time from onset to randomisation), and the source trial added as a random effect to account for clustering. For completeness, analyses will also be performed unadjusted. Results from binary and ordinal analyses (eg, death, dichotomised outcomes of mRS scale, neurological deterioration, SAEs, full-scale mRS) will be presented as ORs with 95% CIs. The proportional odds assumption will be checked before ordinal analyses of outcomes on the mRS are undertaken. If the proportional odds assumption is not met, various standard binary cut-points for poor outcome on the mRS will be used. Continuous or pseudocontinuous outcome analysis results will be presented as mean difference with 95% CI. For time-to-event outcomes, a Cox proportional hazards model will be used to determine HRs with the source trial added as a random effect, and the assumption of proportional hazards will be tested.

Additional analyses will be performed in prespecified subgroups, with an interaction term in models to test heterogeneity.

Sensitivity analyses using a two-stage approach will be conducted to test the robustness of our primary results and will permit (1) inclusion of aggregate data from studies where IPD cannot be obtained to address the issue of data availability bias[27] and (2) assessment with and without studies deemed to have high risk of bias.

An assessment of heterogeneity will be performed before data pooling using the Cochrane Q statistic and $I^2$ statistic.

An assessment of publication bias will be made by visual inspection for funnel plot asymmetry (with and without studies where IPD is obtained) and with Egger's regression test. The Grades of Recommendation, Assessment, Development and Evaluation[28] will be used to evaluate the quality of the synthesised evidence.

### ETHICS AND DISSEMINATION

Ethical approval for the original studies was sought and is documented elsewhere. No new patient data will be collected nor is there any deviation from the original purposes of each study; therefore, further ethical approval is not required. Results from this study will be published in international peer-reviewed journals. All publications from this work will be in the name of the BASC Investigators.

### DISCUSSION

The proposed study will use pooled IPD from RCTs to address the role of BP management in the acute phase of ICH. Conflicting results from multiple clinical trials indicate that BP reduction is not a straightforward question of whether to treat or not. Rather, the problem is complex and needs to take account of patient characteristics, physiological parameters including baseline BP, stroke severity, the timing of treatment, strategy and class of antihypertensive agent, route of administration and dose or target BP.

Elevated baseline systolic BP is associated with haematoma expansion,[29] perihaematomal oedema formation[30] and increased case fatality.[31] There is evidence from some studies that intensive lowering of BP reduces haematoma enlargement,[32] is safe and tolerable[2 3] and does not alter cerebral blood flow.[33] Current guidelines recommend lowering BP early in the course after ICH and that targeting a goal of systolic BP <140 mm Hg is probably safe in patients presenting with an SBP of 150–220 mm Hg.[34 35] Nevertheless, other studies reported no effect of BP reduction on haematoma enlargement,[3 4] and the neutral findings of the ATACH-II trial[4] emphasise that all published data need to be appraised together.

Although guidelines recognise the need for very early treatment, none address the role of prehospital BP reduction. Further evidence is also required to confirm associations between key haemodynamic variables, such as BP variability,[23] with poor outcome. Furthermore, the influence of patient's characteristics that are present prior to ICH on BP lowering interventions is unknown, although these same variables are known to worsen long-term recovery; these include having a prior stroke, white matter lesions or brain atrophy. The proposed investigation aims to provide definitive evidence on these important issues.

### Author affiliations

[1]Centre for Clinical Brain Sciences, University of Edinburgh, Edinburgh, UK
[2]George Institute for Global Health, Faculty of Medicine, University of New South Wales, Sydney, New South Wales, Australia
[3]Stroke Trials Unit, University of Nottingham, Nottingham, UK
[4]National University of Malaysia, Bangi, Malaysia
[5]Neurology Department, Royal Prince Alfred Hospital, Camperdown, New South Wales, Australia
[6]Stroke, Nottingham University Hospitals NHS Trust, Nottingham, UK
[7]Department of Cardiovascular Sciences, University of Leicester, Leicester, UK
[8]National Institute for Health Research Leicester Biomedical Research Centre, Leicester, UK
[9]Department of Internal Medicine, Oslo University Hospital, Oslo, Norway
[10]Neurology Department, Oslo University Hospital, Oslo, Norway
[11]Research and Development, Norwegian Air Ambulance Foundation, Bodo, Norway

[12]Neurology Department, Royal Prince Alfred Hospital, Camperdown, New South Wales, Australia

**Collaborators** This protocol paper is written on behalf of the Blood pressure in Acute Stroke Collaboration (BASC, convener PMB). BASC-ICH comprises the following: ATACH-II: Adnan I Qureshi, Yuko Palesch; CHHIPS: John Potter, ENOS: Philip M Bath, Nikola Sprigg, Joanna M Wardlaw; FAST-Mag: Jeff Saver, Nerses Sanossian; GTN-1/2/3, RIGHT: Philip M Bath; ICH-ADAPT: Ken Butcher; IMAGES: Kennedy R Lees, Keith W Muir; INTERACT: Craig S Anderson; INTERACT2: Craig S Anderson, Hisatomi Arima; PILFAST: Gary Ford; RIGHT-2: Philip M Bath, Nikola Sprigg, Joanna M Wardlaw; SCAST: Eivind Berge, Else C Sandset; VENUS: Janneke Horn.

**Contributors** PMB, TJM, ECS and XW were responsible for first draft of the protocol, study conception and critical review of the current protocol. XW and LJW were responsible for statistical oversight. ZKL, CD, NS, KK, TGR, JW, RASS, EB and CSA were responsible for study conception and critical review of the current protocol.

**Funding** Blood pressure in Acute Stroke Collaboration has not received specific grants from any funding agency in the public, commercial or not-for-profit sectors. Potential trials for inclusion have reported their own funding. TJM is a British Heart Foundation clinical research training fellow. PMB is a Stroke Association Professor of Stroke Medicine. PMB and TGR are NIHR Senior Investigators. CSA is a Senior Research Fellow of the National Health and Medical Research Council of Australia.

**Competing interests** None declared.

**Patient consent for publication** Not required.

**Provenance and peer review** Not commissioned; externally peer reviewed.

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
