## [Reviewer comments · BMJ Open]

ARTICLE DETAILS

TITLE (PROVISIONAL)	Lowering blood pressure after acute intracerebral haemorrhage: protocol for a systematic review and meta-analysis using individual patient data from randomised controlled trials participating in the Blood pressure in Acute Stroke Collaboration (BASC)
AUTHORS	Moullaali, Tom; Wang, Xia; Woodhouse, Lisa; Law, Zhe Kang; Delcourt, Candice; Sprigg, Nikola; Krishnan, Kailash; Robinson, Thompson; Wardlaw, Joanna; Salman, Rustam; Berge, Eivind; Sandset, Else; Anderson, Craig; Bath, Philip

VERSION 1 - REVIEW

REVIEWER	Hao-min Cheng Taipei Veterans General Hospital, Taipei, Taiwan
REVIEW RETURNED	25-Mar-2019

GENERAL COMMENTS	The study protocol is designed to conduct an IPD meta-analysis to address the influence of different blood pressure reduction strategies in patients with acute stroke including ICH! The investigators' efforts should be commended for this important and significant work! The authors, however, might take the following to improve their work! Major 1. The study population should be more clearly defined. Throughout the manuscript, including the introduction and discussion, the IPD meta-analysis is prepared to investigate the patients with acute ICH. However, in the abstract and eligibility criteria, the authors stated that patients with acute stroke including ICH will be included. Whether acute ischemic stroke patients are the candidates of the present study? If yes, some details regarding the background of BP reduction in acute ischemic stroke should be provided and the inclusion criteria should be revised accordingly. If no, the protocol should be specify the study population more clearly2. The research question is not very clearly stated. The current debates are mainly how low and how fast we should lower patients' blood pressure in acute stroke! The research question should be clearly described in the protocol. Minor
--

	1. It is worth noting that the conflicting results of recent studies might be related to different study population characteristics and BP lowering strategies. As such, the relating factors of these possible sources of heterogeneity should be collected from each study. 2. Abstract, line 37: not clear what is one-stage approach 3. Abstract, line 38: with individual patient data, meta-regression is probably not required.
--	---

REVIEWER	Prof. Dr. Simona Lattanzi Marche Polytechnic University, Ancona, Italy
REVIEW RETURNED	03-Apr-2019

GENERAL COMMENTS	This is the description of a protocol for an individual level meta-analysis of randomised controlled trials of blood pressure management in patients with acute spontaneous intracerebral hemorrhage enrolled within the first 7 days of symptom onset. The protocol is well-described and the aim of the study is highly relevant from the clinical point of view. There are, however, that should be further addressed. Systematic reviews and meta-analyses on the same topic have been already performed (Ref. How Should We Lower Blood Pressure after Cerebral Hemorrhage? A Systematic Review and Meta-Analysis. Cerebrovasc Dis. 2017; Intensive blood pressure reduction in acute intracerebral hemorrhage: a meta-analysis. Neurology. 2014). It would be useful to highlight the advantages of a meta-analysis using individual level data and what it could add in comparison to a meta-analysis using aggregate data. In the protocol, it is said that treatment-related haemodynamic parameters will be taken into account. Do you refer to variables of blood pressure variability (BPV)? Many different variables to estimate BPV have been developed and used, including standard deviation, maximum-minimum range, coefficient of variation, variation independent of mean, successive variation, average real variability (ref. Blood Pressure Variability and Clinical Outcome in Patients with Acute Intracerebral Hemorrhage. J Stroke Cerebrovasc Dis. 2015). Please, clarify which variables will be considered and their formulas.
--

VERSION 1 – AUTHOR RESPONSE

Reviewer(s)' Comments to Author:

Reviewer: 1

The study protocol is designed to conduct an IPD meta-analysis to address the influence of different blood pressure reduction strategies in patients with acute stroke including ICH! The investigators' efforts should be commended for this important and significant work! The authors, however, might take the following to improve their work!

Major

1. The study population should be more clearly defined. Throughout the manuscript, including the introduction and discussion, the IPD meta-analysis is prepared to investigate the patients with acute ICH. However, in the abstract and eligibility criteria, the authors stated that patients with acute stroke including ICH will be included. Whether acute ischemic stroke patients are the candidates of the present study? If yes, some details regarding the background of BP reduction in acute ischemic stroke should be provided and the inclusion criteria should be revised accordingly. If no, the protocol should be specify the study population more clearly

Response: We intend to analyse only IPD from cases of acute spontaneous ICH; where RCTs involve both ischaemic stroke and ICH, we will extract IPD from ICH cases only. We include the following statement in the Methods to clarify this point: 'Adults (age >18 years) with spontaneous ICH will be included. For trials including both ICH and ischaemic stroke, we will include only patients with ICH. Participants with secondary ICH (e.g. due to trauma, tumour or vascular malformation) will be excluded.'

2. The research question is not very clearly stated. The current debates are mainly how low and how fast we should lower patients' blood pressure in acute stroke! The research question should be clearly described in the protocol.

Response: We agree that these points are important, and have made the following statements in our manuscript to address these.

Introduction (p5):

'Secondary objectives

- i) Determine the effect of BP lowering according to baseline patient clinical and neuroradiological characteristics, BP lowering strategy (i.e. agent used, or BP target), and timing of the intervention.'
- ii) Determine the effect of BP lowering on radiological outcomes.
- iii) Where sufficient data are available, report associations of on-treatment haemodynamic parameters, adjusted for known confounders.'

Regarding 'how fast?'/timing:

Methods and Analysis (p9, subheading 'subgroup analyses'): 'Where adequate data are available, heterogeneity in the effect of BP lowering on outcomes will be assessed in the following subgroups to determine whether any effects of BP lowering are moderated by patient characteristics or type of BP lowering treatment (agent, target, timing, or place of delivery):'

Supplementary Material 3, suggested figures 2 and 5: subgroup analyses will show whether any effects of BP lowering on the primary outcome are moderated by BP treatment strategy, and time to treatment.

Regarding 'how low?'

Methods and Analysis (p9, subheading 'associations of calculated BP parameters and outcomes'; Supplementary Material 3, table 3): we will show the effect of BP lowering on haemodynamic variables which include mean BP and delta BP (magnitude of BP reduction) and will report associations of these variables in secondary analyses

Minor

1. It is worth noting that the conflicting results of recent studies might be related to different study population characteristics and BP lowering strategies. As such, the relating factors of these possible sources of heterogeneity should be collected from each study.

Response: We agree with this important point and will collect study population characteristics that include demographics (including country), medical history, medication use, and baseline clinical and neuroimaging characteristics, as outlined in Methods and Analysis (p8, subheading 'data items'), and in more detail in Supplementary Material 2.

2. Abstract, line 37: not clear what is one-stage approach

Response: We pre-specify a one stage approach to meta-analysis for our primary analyses as it involves pooling all IPD together and analysing it as if it belongs to one study, with adjustment for confounding variables, and with the source trial as a random effect to account for clustering. A two-stage approach involves analysing IPD from each study separately in the first stage, then pooling the treatment estimates in the traditional meta-analysis way in the second stage – we plan to do this in a sensitivity analysis to test the robustness of our findings. We include more detail about these issues in Methods and Analysis (p10, subheading 'statistical analyses'), and to avoid confusion, have removed reference to these aspects in the Abstract.

3. Abstract, line 38: with individual patient data, meta-regression is probably not required.

Response: We sought further advice on this point, and agree that meta-regression does not make full use of IPD, as it estimates interactions at a study-level, rather than IPD level. We have removed reference to meta-regression from the manuscript, and have amended the title for figure 5 (Supplementary Material 3) as follows:

'Individual participant data regression: relationship between randomised treatment group and odds ratio for unfavourable shift in scores on the modified Rankin Scale at end of trial and (i) time to randomisation; (ii) baseline systolic blood pressure; and (iii) baseline haematoma volume. Model (i) is adjusted for age, sex, severity, type of treatment, and source trial; models (ii) and (iii) are adjusted for the same plus time to randomisation.'

Reviewer: 2

This is the description of a protocol for an individual level meta-analysis of randomised controlled trials of blood pressure management in patients with acute spontaneous intracerebral hemorrhage

enrolled within the first 7 days of symptom onset. The protocol is well-described and the aim of the study is highly relevant from the clinical point of view. There are, however, that should be further addressed.

Systematic reviews and meta-analyses on the same topic have been already performed (Ref. How Should We Lower Blood Pressure after Cerebral Hemorrhage? A Systematic Review and Meta-Analysis. *Cerebrovasc Dis.* 2017; Intensive blood pressure reduction in acute intracerebral hemorrhage: a meta-analysis. *Neurology.* 2014). It would be useful to highlight the advantages of a meta-analysis using individual level data and what it could add in comparison to a meta-analysis using aggregate data.

Response: We have amended Introduction (p 5) in response to this important point:

'IPD meta-analysis is considered the gold-standard for synthesising evidence from RCTs[16] and in this context, provides added value to meta-analyses of aggregate data[17] by facilitating multivariable analyses of treatment effects and subgroup analyses according to time to treatment, treatment strategy used, and baseline characteristics including neurological severity and haematoma volume, adjusted for confounding factors.'

In the protocol, it is said that treatment-related haemodynamic parameters will be taken into account. Do you refer to variables of blood pressure variability (BPV)? Many different variables to estimate BPV have been developed and used, including standard deviation, maximum-minimum range, coefficient of variation, variation independent of mean, successive variation, average real variability (ref. Blood Pressure Variability and Clinical Outcome in Patients with Acute Intracerebral Hemorrhage. *J Stroke Cerebrovasc Dis.* 2015). Please, clarify which variables will be considered and their formulas.

Response: We are familiar with the growing body of literature around BPV and its relation to prognosis after ICH, and of the limitations in assessing BPV across a range of trials where there is likely to be variation in how and when BP was measured during treatment. We will use standard deviation as the primary measure of variability of haemodynamic variables including BP (Supplementary Material 3, Table 3), as this is probably the most accessible index for clinicians and appears to be the strongest prognostic variable. However, we appreciate the limitations of this approach when applied to analyses of associations of variability and outcome (e.g. confounding with mean) and secondary analyses of associations of variability will include various indices of variability to test the robustness of our findings. However, such analyses of associations of post-randomisation haemodynamic data are post-hoc/observational, and secondary to the primary aims of this meta-analysis, therefore we provide limited detail in this manuscript (as below):

'Once all data have been pooled, an assessment of the new dataset will be made with a view to use trial BP and heart rate measures to calculate important BP lowering haemodynamic parameters, a number of which have been associated with outcome after ICH.[23–26] The aim is to determine the prognostic significance of these calculated variables, adjusted for all known confounders, and to present these data in secondary analyses.'

VERSION 2 – REVIEW

REVIEWER	Hao-min Cheng Taipei Veterans General Hospital, Taipei, Taiwan
REVIEW RETURNED	07-May-2019

GENERAL COMMENTS	I thank the authors for the considered responses to reviewers' questions. My initial concerns have been addressed and I have no other issues.
---

REVIEWER	Simona Lattanzi Marche Polytechnic University, Italy
REVIEW RETURNED	03-May-2019

GENERAL COMMENTS	The Authors addressed the queries.
------------------------------------